# Truncated Analogues of a G-Quadruplex-Forming Aptamer Targeting Mutant Huntingtin: Shorter Is Better!

**DOI:** 10.3390/ijms232012412

**Published:** 2022-10-17

**Authors:** Claudia Riccardi, Federica D’Aria, Dominga Fasano, Filomena Anna Digilio, Maria Rosaria Carillo, Jussara Amato, Laura De Rosa, Simona Paladino, Mariarosa Anna Beatrice Melone, Daniela Montesarchio, Concetta Giancola

**Affiliations:** 1Department of Chemical Sciences, University of Naples Federico II, 80126 Naples, Italy; 2Department of Pharmacy, University of Naples Federico II, 80131 Naples, Italy; 3Department of Advanced Medical and Surgical Sciences, 2nd Division of Neurology, Center for Rare Diseases and InterUniversity Center for Research in Neurosciences, University of Campania Luigi Vanvitelli, 80131 Naples, Italy; 4Department of Molecular Medicine and Medical Biotechnology, University of Naples Federico II, 80131 Naples, Italy; 5Research Institute on Terrestrial Ecosystems (IRET), UOS Naples-CNR, 80131 Naples, Italy; 6Department of Experimental Medicine, University of Campania Luigi Vanvitelli, 80138 Naples, Italy; 7Sbarro Institute for Cancer Research and Molecular Medicine, Center for Biotechnology, Temple University, Philadelphia, PA 19122-6078, USA

**Keywords:** G-quadruplex, aptamers, physicochemical characterization, Huntington’s disease, *Drosophila melanogaster* model

## Abstract

Two analogues of the MS3 aptamer, which was previously shown to have an exquisite capability to selectively bind and modulate the activity of mutant huntingtin (mHTT), have been here designed and evaluated in their physicochemical and biological properties. Featured by a distinctive propensity to form complex G-quadruplex structures, including large multimeric aggregates, the original 36-mer MS3 has been truncated to give a 33-mer (here named MS3-33) and a 17-mer (here named MS3-17). A combined use of different techniques (UV, CD, DSC, gel electrophoresis) allowed a detailed physicochemical characterization of these novel G-quadruplex-forming aptamers, tested in vitro on SH-SY5Y cells and in vivo on a *Drosophila* Huntington’s disease model, in which these shorter MS3-derived oligonucleotides proved to have improved bioactivity in comparison with the parent aptamer.

## 1. Introduction

Huntington’s disease (HD) is an autosomal dominant and progressive neurodegenerative disorder with a distinct phenotype, including motor symptoms, such as chorea, dystonia, and incoordination, associated with psychiatric signs, such as depression and anxiety, cognitive decline, and behavioral difficulties [1,2,3,4,5]. Although HD has received and continuously receives a great deal of research attention, unfortunately, no definitive treatment is currently available to recover from this invalidating disease. Some HD manifestations well respond to symptomatic therapies, but for many other symptoms, effective treatments are not accessible [6,7,8,9].

This debilitating disorder is mainly caused by a CAG trinucleotide repeat in the gene encoding for the wild-type huntingtin (HTT) protein. The expansion of the CAG repeats is translated into a mutant huntingtin (mHTT) protein with an abnormally long polyglutamine strand at the N-terminus [10,11]. Evidence suggests that this long, additional tail confers different functions to mHTT compared with the normal protein [12,13,14,15], and these differences could be suitably exploited to selectively target and modulate the mutant protein for an effective therapy.

In this context, highly specific nucleic-acid-based aptamers—that is, short, single-stranded DNA or RNA molecules generally identified from a large pool of random oligonucleotide sequences by SELEX (systematic evolution of ligands by exponential enrichment) [16,17] and proposed as valuable therapeutic agents in different diseases [18,19,20,21,22]—have been selected for their activity in neurodegenerative disorders [23,24,25,26,27], and some examples have been recently reported also for HD treatment [28,29,30,31,32,33,34,35].

In 2018, Shin and colleagues identified by SELEX a set of four guanine-rich DNA-based aptamers (MS1, MS2, MS3, MS4), able to selectively recognize the C-terminal domain of mHTT, with an expanded 78-residue polyglutamine tract, but not the wild-type protein. Among these aptamers, MS3, the 36-mer with the sequence 5′-GGGA(GGGA)_8_-3′, proved to be the strongest mHTT binder [34]. Aiming at providing a deep insight into the physicochemical and biological features of MS3, we recently investigated the conformational behavior of this aptamer, along with its nuclease resistance in vitro and biological activity in vitro and in vivo [36].

Our results demonstrated that MS3 adopts a very stable parallel G-quadruplex (G4) structure, as determined by UV spectroscopy (UV), circular dichroism (CD), and differential scanning calorimetry (DSC) analyses, and shows high resistance to nuclease digestion in pseudo-physiological solutions [36]. Then, confocal microscope analysis proved that fluorescein-labelled MS3 is rapidly internalized in both non-neuronal HeLa and neuronal SH-SY5Y cells in a dose-dependent manner and persists in the examined neuronal cell lines up to 72 h, with no significant cytotoxicity [36]. Finally, using a well-established *Drosophila melanogaster* model for HD (Q128HD-FL), which expresses the mHTT protein, we proved that the neuronal function improved after MS3 treatment, definitively demonstrating the in vivo efficacy of this aptamer [36].

These findings are overall very promising and stimulated further optimization studies on this aptamer, needed to improve and regulate its functions to allow more advanced in vivo studies.

Among the different post-SELEX optimization strategies, such as dimerization/multimerization [20,37] and chemical modification [21,38,39], rationally designed truncation can be a very simple and powerful approach to improve aptamer binding affinity and therapeutic efficacy [40,41,42]. Indeed, truncation is intended to specifically remove the nucleotides not strictly necessary for the interaction with the target. In many cases, this approach can be effective in reducing the conformational polymorphism of the parent aptamer, thus favoring a well-defined folding. As an additional advantage, shorter oligonucleotide sequences are produced in higher yields, more rapidly and with lower synthetic costs [40,42]. In this scenario, efforts to obtain minimized oligonucleotide sequences that possess the same or better performance compared with the original full-length aptamers are indeed necessary to advance this research field from the bench to the bedside.

Thus, aiming at optimizing the MS3 aptamer through the identification of improved analogues, we here designed and investigated in detail two truncated MS3-derived oligonucleotides, analyzed in their physicochemical and biological behavior.

## 2. Results and Discussion

### 2.1. MS3 Forms Large Aggregates: Design of Truncated MS3 Analogues to Reduce Its Heterogeneity

To guide the design of optimized variants of MS3, this aptamer was analyzed by polyacrylamide gel electrophoresis (PAGE) under native conditions to preliminarily verify how many species this oligonucleotide is able to form in different pseudo-physiological saline conditions (Figure 1).

Native gel electrophoresis analysis allows a general characterization of the migration ability of different oligonucleotides, providing precious information on their molecularity. For these studies, we tested different saline conditions: two Na^+^-rich buffer solutions, indicated as Na^+^ buffer (10 mM NaH_2_PO_4_/Na_2_HPO_4_, 90 mM NaCl, pH = 7.0) and PBS (137 mM NaCl, 2.7 mM KCl, 10 mM NaH_2_PO_4_/Na_2_HPO_4_, 1.8 mM KH_2_PO_4_/K_2_HPO_4_, pH = 7.4), along with a K^+^-rich one (10 mM KH_2_PO_4_/K_2_HPO_4_, 70 mM KCl, 0.2 mM EDTA, pH = 7.3).

To better define the formed species, we also included in the gel an additional reference oligonucleotide, that is, the 25-mer V7t1 aptamer (Figure 1, lane 4), previously demonstrated to fold into both dimeric and monomeric species, with a net predominance of dimeric forms when not subjected to thermal treatments (not-annealed samples, N.A.) [43].

Under all the explored conditions, independently from the buffer solution used, the 36-mer MS3 showed multiple bands, presumably of monomeric and dimeric species, which were slightly retarded compared with the monomeric and dimeric species of the 25-mer V7t1 due to its higher length.

These monomeric and dimeric forms of MS3 were accompanied also by very large species (aggregates or higher-order G4 structures), which migrated as smeared bands in the gel and hardly entered the polyacrylamide pockets (Figure 1, lanes 1–3), as also previously observed for other G-quadruplex aptamers forming large aggregates in a solution [44,45].

Aiming at reducing the formation of large aggregates and simplifying the conformational polymorphism of MS3, we thus designed two truncated versions of this 36-mer of the sequence 5′-GGGA(GGGA)_8_-3′, however, maintaining the central core of the original oligonucleotide, which guarantees a distinctive G-quadruplex folding capability. In detail, we selected: (i) a 33-mer, obtained by removing the 5′-terminal GGG trinucleotide in MS3 sequence (here named MS3-33, of the sequence 5′-A(GGGA)_8_-3′), which could favor end stacking and thus spontaneous aggregation between multiple G-quadruplex-structured molecules, and (ii) a 17-mer (here named MS3-17, of the sequence 5′-A(GGGA)_4_-3′), whose length is approximately half that of MS3-33, not containing terminal contiguous 2′-deoxyguanosines and including the minimal G-rich motif, common to both MS3 and MS3-33, able to provide a three tetrad-based G-quadruplex folding.

In analogy to our previous study [36], these oligonucleotides were thus investigated in their physicochemical behavior in pseudo-physiological buffers differing for the metal cation type and/or concentration to evaluate the peculiar conformations they adopt in the presence of different metal cations and/or ionic strength [46,47,48]. These analyses were carried out in the same saline conditions we previously used for the MS3 characterization, that is, in a buffer exclusively containing sodium ions as cations (10 mM NaH_2_PO_4_/Na_2_HPO_4_, 90 mM NaCl, pH = 7.0, here indicated as Na^+^ buffer) and in another containing small amounts of potassium ions (PBS, i.e., 137 mM NaCl, 2.7 mM KCl, 10 mM NaH_2_PO_4_/Na_2_HPO_4_, 1.8 mM KH_2_PO_4_/K_2_HPO_4_, pH = 7.4), to mimic different pseudo-physiological conditions.

### 2.2. UV Spectroscopy Characterization of the MS3 Analogues: UV Thermal Difference Spectra and UV Thermal Denaturation/Renaturation Experiments

First, we recorded UV thermal difference spectra (TDS, Appendix A) of the MS3 analogues at 2 μM oligonucleotide conc., comparing them with those previously obtained for their parent aptamer [36].

The UV spectral difference registered for the unfolded and folded (at high and low temperature, i.e., 100 and 15 °C, respectively) oligonucleotide represents indeed a diagnostic “fingerprint” of a specific nucleic acid secondary structure and is therefore very useful to verify whether, in the explored saline conditions, a G-rich sequence effectively adopts a G-quadruplex folding [49].

In analogy with the behavior previously found for MS3 [36], the normalized TDS profiles of MS3-33 and MS3-17 fully confirmed their folding in G-quadruplex structures, showing in all cases two positive (at ca. 240 and 275 nm) and two negative (around 260 and 295 nm) bands (Appendix A) [49].

Then, we also determined their TDS factors for an estimation of the predominant G-quadruplex conformation in a solution [50]. In both the analyzed saline conditions, the calculated ΔA_240_/ΔA_295_, ΔA_255_/ΔA_295_, and ΔA_275_/ΔA_295_ factors for MS3-33, similar to MS3 [36], provided values higher than 4, 3.5, and 3, respectively (Appendix A), which were all indicative of a parallel G-quadruplex folding, in accordance with literature data [50]. In turn, in both buffer solutions, MS3-17 provided TDS factor values that cannot be unambiguously associated with a single, well-defined G-quadruplex topology and indeed suggest the coexistence of both parallel and antiparallel, or alternatively parallel and hybrid, G-quadruplex structures (in fact, values lower than 2, 1.5, and 2, respectively, for ΔA_240_/ΔA_295_, ΔA_255_/ΔA_295_, and ΔA_275_/ΔA_295_ are indicative of antiparallel or hybrid G4 folding) [50].

UV melting/cooling experiments were then performed at 2 µM oligonucleotide conc. in the 15–90 °C temperature range following the UV signal at 295 nm. For both MS3-33 and MS3-17, a nice sigmoidal decrease of the absorbance at 295 nm was found upon increasing the temperature in the 45–70 °C range for MS3-33, and 35–70 °C for MS3-17 (Appendix A, panels a and b, respectively, for PBS and Na^+^ buffer), in agreement with the expected thermal denaturation of their G-quadruplex structures [51,52].

For both aptamers, the UV analysis performed in PBS and Na^+^ buffer showed almost superimposable melting/cooling profiles and only limited hysteresis (Appendix A, respectively, for MS3-33 and MS3-17), indicating that under the experimental conditions tested (scan rate = 1 °C min^−1^), the related denaturation/renaturation processes were essentially reversible, except for MS3-17 in Na^+^ buffer, where a significant hysteresis was detected.

Notably, for both MS3 analogues, the calculated apparent *T_m_* values (Table 1) were higher in PBS than in the Na^+^ buffer (apparent *T_m_* values of 63 and 59 °C for MS3-33 and 52 and 46 °C for MS3-17, respectively), indicating a higher stability of the G-quadruplex structures in the first buffer, well explained by the better ability of K^+^ to stabilize G-quadruplex structures than Na^+^ ions [46,47,48] and the overall higher ionic strength of PBS vs. the selected Na^+^ buffer.

### 2.3. CD Spectroscopy Characterization of the MS3 Analogues: CD Spectra and CD Thermal Denaturation/Renaturation Experiments

CD experiments were carried out to study the conformational properties and thermal stability of the G4 structures of the here studied MS3 analogues [53]. In both selected buffer solutions, CD spectra of MS3-33 and MS3-17, collected at 5 °C, showed the characteristic profile of a predominantly parallel G4 folding, with a maximum at 263 nm and a minimum at 240 nm (Figure 2) [54,55]. The positive CD bands showed higher intensities in PBS than in the Na^+^ buffer, suggesting a higher structuration degree in PBS, in line with the higher stability of G4s in K^+^-containing solutions [46,47,48].

In order to better investigate the G4 structure topology adopted by MS3-33 and MS3-17 in the examined solutions, CD spectra recorded at 5 °C were also processed by singular value decomposition (SVD) analysis, using a software developed by del Villar-Guerra et al. [56]. This SVD analysis confirmed for MS3-33 the exclusive formation of a parallel G4 topology in both buffers (Appendix A), in accordance with the determined TDS factor values (Appendix A) and similar to MS3 [36]. In turn, SVD analysis of the CD spectra of MS3-17 indicated that in PBS buffer, this oligonucleotide essentially adopted a parallel G4 folding, whereas in the Na^+^ buffer, it was present as a mixture of both parallel and hybrid G4 structures (Appendix A, 70% vs. 30%), in line with the calculated TDS factor values (Appendix A) previously discussed in Section 2.2.

CD melting and cooling experiments were carried out on both MS3 analogues dissolved in the Na^+^ buffer (Figure 3a,b) and in PBS (Appendix A), following the CD signal changes at 263 nm as a function of temperature in the 5–100 °C range at the scan rate of 0.5 °C min^−1^. The melting and cooling curves were almost superimposable in both buffer conditions, indicating a reversible heating and cooling process. Notably, in these buffers, the CD spectra of MS3-33 and MS3-17 at 5 °C before and after a melting/cooling cycle were basically superimposable (Figure 2, black and blue lines, respectively), further confirming the reversibility of the unfolding/refolding processes. Interestingly, while in the Na^+^ buffer, the two MS3 analogues were fully denatured at 100 °C (Figure 2b,d, red line), in PBS, full denaturation was not achieved even at 100 °C (Figure 2a,c, red line), although the initial spectral features were completely recovered after the melting/cooling process. Moreover, the melting curves showed similar features if recorded at a scan rate of 1 °C min^−1^, indicating that, in these solution conditions, the overall folding–unfolding process for both oligonucleotides was not kinetically controlled (Appendix A). The melting temperatures obtained by the CD melting curves of MS3-33 and MS3-17 in both selected buffers are reported in Table 2.

CD data previously reported for MS3 in the Na^+^ buffer (*T_m_* = 50 °C) [36] show that, in the same solution conditions, MS3-33 adopts a G4 structure with comparable stability as the parent aptamer, whereas MS3-17 gives less stable G4 structures (Appendix A).

The inspection of data reported in Table 2 shows that the CD-derived *T_m_* values of both MS3 analogues in PBS are in good agreement with those obtained by UV melting experiments, whereas the *T_m_* values obtained in the Na^+^ buffer clearly differ from the UV-derived data. This discrepancy can be attributed to the presence of different species, which are differently highlighted, monitoring the melting process at two different wavelengths for the two spectroscopic techniques (λ = 295 nm for the UV melting and λ = 263 nm for the CD melting). This is an indication that both of the G4-forming oligonucleotides can assemble into dimeric or multimeric structures, as also revealed by the gel electrophoresis experiments (see below, Section 2.5). Then, for a deeper insight into the thermal stability of the G-quadruplex structures of the MS3 analogues, DSC measurements were performed (see below, Section 2.4).

### 2.4. Differential Scanning Calorimetry (DSC) Experiments

DSC is the methodology of choice to study the energetics of G4 melting, providing information on the stability and structural complexity of these non canonical nucleic acid structures [57,58]. Calorimetric profiles of MS3-33 and MS3-17 in PBS and Na^+^ buffer are shown in Figure 4, and the relative thermodynamic parameters are reported in Table 3. DSC curves of two consecutive melting experiments are superimposable, confirming the reversibility of folding/unfolding process. All the melting temperatures proved to be in good agreement with those extracted by UV melting curves and, in the case of the analyses in PBS, also by CD melting measurements. The DSC curves of MS3-33 and MS3-17 in both selected buffers were also fitted with a two-states model by the van’t Hoff analysis. Interestingly, calorimetric enthalpies, Δ*_exp_H°*, were lower than Δ*_vH_H°*. As well described in the literature, the condition Δ*_exp_H°* < Δ*_vH_H°* can be explained by the presence of aggregates of G4 structures, forming dimeric or multimeric species, which in the calorimetric enthalpy determination leads to underestimate the effective concentration of the initial species during the melting process [59].

Taken together, the calorimetric and spectroscopic data provide relevant information on the stability and topological arrangement of the two MS3 analogues. DSC data indicated that in the Na^+^ buffer, MS3-33 (*T_m_* = 57 °C) showed higher thermal stability than its parent aptamer, MS3 (*T_m_* = 53 °C), whereas MS3-17 (*T_m_* = 47 °C) was slightly less stable. In PBS, both MS3 analogues showed high thermal stability (*T_m_* values of 60 and 57 °C, respectively) and, considering the DSC profiles at 37 °C, proved to be in their respective fully folded states (Figure 4).

### 2.5. Native Gel Electrophoresis Analysis

PAGE experiments under native conditions were performed to characterize the MS3 analogues and determine the number of species they form in a solution. In Figure 5, a representative example of a 10% polyacrylamide gel—in which the MS3 analogues were compared with their parent aptamer and the 25-mer V7t1 oligonucleotide used as control [43]—is reported.

Similar to MS3 (Figure 5, lanes 2 and 5, respectively, for PBS and Na^+^ buffer), MS3-33 formed multiple bands of monomeric and dimeric species, along with large species (aggregates or higher-order G4 structures), which migrated as smeared bands in the gel and hardly entered the polyacrylamide pockets (Figure 5, lanes 3 and 6, respectively, for PBS and Na^+^ buffer).

Interestingly, MS3-17 behaved differently, not forming large aggregates under the explored conditions, but only multiple monomeric and dimeric forms, plausibly present as distinct conformations. Both the monomeric and dimeric species of MS3-17 migrated faster than the corresponding V7t1 forms, as expected for a 17-mer compared with the anti-VEGF 25-mer.

Notably, for all the investigated aptamers, no marked difference in the number of formed species and/or in their migration ability was found on varying the tested saline conditions.

### 2.6. MS3 Analogues Are Resistant to Nuclease Degradation

A crucial parameter for an effective in vivo use of aptamers is their resistance to nucleases, which can rapidly degrade oligonucleotides in the cellular environments, thus limiting their in vivo activity.

Aiming at evaluating the stability of MS3-33 and MS3-17 toward the enzymatic digestion, both oligonucleotides were incubated in 80% (*v*/*v*) fetal bovine serum (FBS) at 37 °C, and their integrity was monitored up to 48 h. Samples withdrawn from these reaction mixtures at fixed times, as indicated in Figure 6 and Appendix A, were then analyzed by gel electrophoresis under denaturing conditions (Figure 6) following previously described procedures [60,61]. The intensity of each oligonucleotide band on the gel was then calculated and expressed as a normalized percentage with respect to that of the corresponding untreated oligonucleotide (Appendix A).

PAGE experiments revealed for both oligonucleotides a slow reduction in the DNA band intensity up to 6 h, where the remaining intact sequence was ca. 30% of the initial amount for MS3-33 and around 15% for MS3-17 (Figure 6 and Appendix A). In the subsequent time monitoring, MS3-33 still maintained about 30% of its initial amount, whereas MS3-17 further decreased in intensity (only ca. 5%–3% of it was present at 30 and 48 h, Appendix A), as a consequence of the nuclease degradation.

An additional band with faster mobility, attributable to shorter oligonucleotide species, which are expected to form in serum, was observed after 30 min and 1 h, respectively, for MS3-17 and MS3-33 (Figure 6). Notably, both aptamers still maintained a certain amount of intact form after 24 h incubation in FBS (5% and 30% for MS3-17 and MS3-33, respectively), in no case being completely degraded in the explored 48 h time.

Overall, even if not reaching the nuclease resistance of their MS3 precursor, for which ca. 50% of the intact oligonucleotide was found after 24 h 80% FBS incubation [36], these aptamers exhibited a good enzymatic stability in the strong tested conditions up to 2 (for MS3-17) or 6 h (for MS3-33), in line with the behavior of oligonucleotides forming stable G-quadruplex structures in a solution [62].

### 2.7. MS3 Analogues Are Efficiently Internalized in a Neuronal Cellular Model and Do Not Display Significant Cytotoxic Properties

To evaluate the biological activity of the here studied MS3 analogues, we first analyzed their cellular uptake ability, in analogy to previous experiments [36], and compared their cellular adsorption properties with those of MS3.

For this aim, neuroblastoma-derived SH-SY5Y cells were incubated for 24 h with different concentrations of fluorescein isothiocyanate (FITC)–conjugated MS3, MS3-33, and MS3-17. Fluorescent signals were detected for all the investigated samples (Figure 7), indicating that they were efficiently internalized. Similar to MS3, the rate of cellular uptake increased in a dose-dependent manner for MS3-33, as clearly shown by quantitative analysis (Figure 7, graph), thus indicating that its uptake was a concentration-dependent process. In contrast, fluorescence intensities for MS3-17 were similar in the 1–8 μM concentration range (Figure 7, graph), pointing at a saturable adsorption of this aptamer. Strikingly, a statistically significant increase of the MS3-17 uptake was detected at 12 μM concentration (Figure 7). Thus, all these data indicate that the two analogues display a different behavior in terms of cellular uptake.

Overall, the fluorescence intensities were lower for both MS3-33 and MS3-17 with respect to MS3 (Figure 7, Appendix A), evidencing that MS3 displayed the highest rate of cellular uptake among the three studied oligonucleotides. It is noteworthy that at the lowest studied concentration, that is, 1 μM, MS3-17 was adsorbed more efficiently than MS3-33, appearing in discrete fluorescent puncta in all cells (Figure 7). The lower efficiency and poor dose-dependent uptake of the two MS3 analogues compared with MS3 could be related to structural effects. Through a systematic structural and biological analysis of five sequence-related DNA molecules, Roxo et al. recently demonstrated that the cellular uptake efficiency of a G-quadruplex-forming oligonucleotide correlates with its length and, as a consequence, with the number of its G-tetrads [63], possibly because of the higher charge neutralization [64]. Interestingly, the shorter G-quadruplex-forming oligonucleotides were more efficient in their antiproliferative activity notwithstanding their lower uptake rate [63].

Furthermore, in light of a recent study showing that the efficiency of the cellular uptake of DNA nanostructures may be modulated by the chemical properties of the attached fluorophore and its position [65], we cannot exclude that the uptake of the unlabeled aptamers may differ from the fluorescein-labeled ones. On the other hand, our data, based on the use of the same fluorophore, pointed out that, among the three studied aptamers, MS3 is the most efficiently adsorbed one by SH-SY5Y cells.

Remarkably, no evident effects of cytotoxicity upon treatment with either MS3-33 or MS3-17 were detected, as evidenced by the regular shape of nuclei stained with DAPI (Figure 7) and by typical cell morphology observed in a phase contrast microscope (similar to MS3 [36]). Moreover, we observed that the percentage of live/dead cells, assessed by trypan blue staining, and their metabolic activity, evaluated by the MTT assay, are comparable in cells treated with the aptamers with respect to untreated cells (Appendix A), further confirming that the studied aptamers exert no general cytotoxicity at the tested μM concentrations.

Since MS3 uptake was time dependent and the aptamer proved to be quickly internalized in the cells as early as 1 h [36], we analyzed the timing of cellular entry for its analogues paying attention to the early time points. For this aim, time-course experiments were performed, incubating SH-SY5Y cells with the FITC-conjugated aptamers at 4 µM, selected for being the best explored concentration with a good uptake rate for all the aptamers (Figure 8).

A valuable fluorescent signal was detected at as early as 0.5 h for all the three samples, which progressively increased over time (Figure 8, bottom graph). Moreover, at very early time points (0.5 and 1 h), the main fluorescent signal is on a plasma membrane and/or in very close proximity for all the aptamers, as evidenced by the typical ring fluorescence pattern resembling the cell boundaries (Figure 8). Interestingly, MS3-17 was rapidly internalized with respect to MS3 and MS3-33: ca. 50% of MS3-17 is in intracellular dots at 2 h monitoring, while a higher amount of MS3 and MS3-33 is still at the plasma membrane at the same time point, as also clearly shown by quantitative analysis (Figure 8, graph on the right). Moreover, MS3-17 proved to be almost completely taken up by the cells (95–96%) at 4 h (Figure 8, graph at right), further supporting its quick and efficient internalization.

To evaluate the stability of MS3 analogues in cells, combined wash-out and time-course experiments were performed. Specifically, SH-SY5Y cells were incubated with the fluorescent aptamers (4 µM) for 24 h (time 0), washed to remove the unincorporated ones, and then incubated for different time periods in the culture medium (Figure 9).

In agreement with what was previously observed for MS3 [36], both MS3 analogues proved to be quite stable inside the cells; indeed, a valuable fluorescent signal was still detectable until 72 and 96 h, although it progressively decreased over time (Figure 9, left graph). As clearly evident from the quantitative analysis (Figure 9, graphs), MS3 and MS3-33 displayed similar kinetics of fluorescence decay, while MS3-17 behaved differently. Interestingly, 50% reduction of the total fluorescent signal was reached after 24 h for MS3 and MS3-33, whereas for MS3-17, the same result was obtained after 48 h. Altogether, these data indicated a good persistence of all the three aptamers, with MS3-17 showing the best performance in the series.

Taken together, our data indicated that both the truncated MS3 analogues could be rapidly and efficiently taken up by cells without inducing any significant cytotoxic effects, supporting their potential use for in vivo therapeutic strategies. Moreover, the quick internalization and higher stability inside the tested cells of FITC-conjugated MS3-17 compared with MS3 and MS3-33 suggested for this aptamer an improved biological activity, notwithstanding its overall lower levels of cell uptake.

### 2.8. The Truncated MS3 Analogues Rescued Neuronal Deficits in a Drosophila Huntington’s Disease Model

To confirm the efficacy of the two truncated MS3 analogues in vivo, both MS3-33 and MS3-17 were tested using a well-established Huntington’s disease Drosophila model (Q128HD-FL) [66,67,68]. These transgenic flies (genotype: elav-Gal4/+; UAS-HttFL-Q128/+) express a pathogenic full-length human cDNA, encoding a mutated HTT protein (mHTT) containing a 128 Q repeat in exon 1. The expression of mHTT in the nervous system of transgenic flies causes a very aggressive course of HD with most of the pathological hallmarks of the disease, including decreased lifespan and age-dependent locomotor dysfunction. To test whether—similar to MS3—MS3-33 and MS3-17 were able to slow down both the course of the disease and the onset of pathological symptoms, adult motor function and survival, as parameters of improved health, were measured. Due to impaired motor neuronal function, HD flies exhibited progressive locomotion disorders that could be assessed by the climbing test, which takes advantage of the strong negative geotaxis behavior of Drosophila.

Transgenic flies were fed with four different foods, a control diet with an assay fly food (AF) medium as control and three AF media supplemented with MS3 or one of its truncated analogues, throughout adulthood, from fly eclosion to death. In this work, a comparative study of all the selected aptamers were carried out. All the aptamers were administered at 6.25 µM concentration, which corresponds to the minimum concentration of MS3 that gave the best result in our previous experiments [36]. Huntington’s flies were tested on the day of eclosion (day 1) and 3, 6, 9, and 12 days after eclosion, and both the percentage of flies climbing over 9 cm and the average climbing height reached were calculated.

Obtained results clearly showed that both MS3-33 and MS3-17 significantly improved the climbing ability of Q128HD-FL transgenic flies (Figure 10a,b). For both aptamers, motor dysfunction appeared later, with a higher percentage of flies that achieved the target (9 s) over time with respect to the MS3-treated flies (**** *p* < 0.0001). Furthermore, as shown in Figure 10b, HD flies treated with both MS3 analogues reached a significantly higher average height even on day 12 after eclosion, compared with HD flies treated with MS3, which, similar to untreated HD flies, tended to remain at the bottom of the vial (**** *p* < 0.0001).

In addition, since HD flies have a short life, the influence of these two novel aptamers on the HD lifespan was evaluated. Obtained results indicated that both aptamers significantly improved the survival rate of Q128HD-FL flies with respect to untreated HD flies. As shown in Figure 10c, the curves of HD flies treated with MS3-33 and MS3-17 were similar and differed significantly compared with untreated HD flies (**** *p* < 0.0001). After an initial overlap period of 5 days, the respective curves diverged due to the higher survival rate of the flies treated with the MS3 analogues (median survival of 15 days for MS3-17 and 14 days for MS3-33 vs. 12 days for untreated flies). The survival curves of both treated groups declined slowly in comparison with the curve of the untreated flies that dropped on day 7. In addition, both aptamers extended the mean lifespan by as much as 34.8% and 31.5%, respectively, and the maximum lifespan by 62.5% and 43.7% compared with untreated flies. Overall, all these data suggest that both aptamers can slow down the progression of the disease.

## 3. Materials and Methods

### 3.1. Materials

Acrylamide/bis-acrylamide (19:1) 40% solution, glycerol, formamide, urea, and GelGreen nucleic acid stain were purchased from VWR (Milan, Italy). Ammonium persulfate (APS) and tetramethylethylenediamine (TEMED) were purchased from Sigma-Aldrich (Merck Life Science, Milan, Italy). Fetal bovine serum (FBS) was provided by Euroclone (Milan, Italy).

### 3.2. Oligonucleotide Sample Preparation

MS3 (5′-GGGA(GGGA)_8_-3′), MS3-33 (5′-A(GGGA)_8_-3′), and MS3-17 (5′-A(GGGA)_4_-3′), as well as their FITC-conjugated derivatives, were purchased from Biomers.net GmbH (Ulm, Germany) as HPLC-purified sequences. All the used oligonucleotides were characterized by MALDI and proved to be >97% pure by HPLC analysis, as provided by the manufacturer.

The concentration of each oligonucleotide was evaluated by UV measurements at 260 nm and 95 °C, using molar extinction coefficient values calculated by the nearest-neighbor model [69]. For the physicochemical characterization, all the oligonucleotides were analyzed either in sodium phosphate buffer (10 mM NaH_2_PO_4_/Na_2_HPO_4_, 90 mM NaCl, pH = 7.0) or in PBS (137 mM NaCl, 2.7 mM KCl, 10 mM NaH_2_PO_4_/Na_2_HPO_4_, 1.8 mM KH_2_PO_4_/K_2_HPO_4_, pH = 7.4). Only for the native PAGE analysis of MS3, also a K^+^-rich buffer solution (i.e., 10 mM KH_2_PO_4_/K_2_HPO_4_, 70 mM KCl, 0.2 mM EDTA, pH = 7.3) was explored. Before use, the solutions were heated at 90 °C for 5 min and then slowly cooled to room temperature.

### 3.3. UV Spectroscopy Analysis

UV spectra and UV thermal denaturation/renaturation measurements were performed on a Jasco V-770 UV–Vis spectrophotometer equipped with a Peltier Thermostat Jasco ETCS-761, by using a quartz cuvette with a 1 cm path length (1 mL internal volume, Hellma). Both MS3-derived oligonucleotides were dissolved in the selected buffer solution to obtain a 2 μM concentration and then slowly annealed before use.

Absorbance spectra were recorded at 15 and 100 °C in the 220–320 nm range using a scanning speed of 100 nm/min with the appropriate baseline subtracted [46,52]. TDS data were obtained by subtracting the UV spectrum recorded at a temperature below the *T_m_* (15 °C), at which the aptamer was fully structured, from the one obtained at a temperature above *T_m_* (100 °C), where the G4 structure was fully denatured [49,52]. Each experiment was performed in duplicate. In order to allow an easy comparison of the spectral data, all the obtained differential spectra were then normalized to the maximum of absorbance simply by dividing the raw data in the 220–320 range by its maximum value, so that the highest positive peak had a Y-value of +1 [49]. From normalized spectra, TDS factors (ΔA_240_/ΔA_295_, ΔA_255_/ΔA_295_, and ΔA_275_/ΔA_295_) were also calculated in both the analyzed saline conditions as the ratios between the absolute absorbance values at different wavelengths [49,50].

The absorbance vs. temperature profiles of both oligonucleotides were monitored following the absorbance changes at 295 nm in the 20–90 °C temperature range with a scan rate of 1 °C min^−1^ [51,52]. Data from UV thermal denaturation/renaturation profiles were also converted into normalized ΔA values (NΔA) as a function of temperature, as previously described [60]. Apparent *T_m_* values were estimated as the maxima of the first derivative plots of the melting/cooling curves, and the error associated with the *T_m_* determination was ±1 °C. Each experiment was performed in duplicate.

### 3.4. Circular Dichroism (CD) Analysis

CD spectra were recorded on a Jasco J-815 spectropolarimeter (Jasco Inc., Tokyo, Japan) equipped with a PTC-423S/15 Peltier temperature controller, using a quartz cuvette with a path length of 1 cm. All the spectra were recorded at 5 °C in the 220–340 nm wavelength range and averaged over three scans. The following parameters were used: 100 nm min^−1^ scan rate, 0.5 s response time, and 1 nm bandwidth. The buffer baseline was subtracted from each spectrum. The sample concentration was 2 µM for all DNA samples. CD melting and cooling experiments were carried out in the 20–100 °C range at 0.5 °C min^−1^ heating rates following changes of the CD signal at the wavelength of maximum intensity (263 nm). Data from CD thermal denaturation/renaturation profiles were also converted into folded fraction, as previously described [62,70]. The melting temperature values (*T*_*m*_) were determined from a curve fit using the Origin 7.0 software (OriginLab Corp., Northampton, MA, USA). Each experiment was performed in duplicate, and the reported values averaged.

### 3.5. Deconvolution of CD Spectra

To perform the SVD analysis, CD spectra acquired at 5 °C were treated to convert the *Y*-axis unit from mdeg to molar ellipticity, as previously described [45]. The resulting spectra were then processed as reported in the literature [56].

### 3.6. Differential Scanning Calorimetry (DSC) Analysis

DSC experiments were performed on a nano-DSC (TA Instruments, New Castle, DE, USA), analyzing the oligonucleotide samples at a 200–250 µM concentration. Three heating/cooling cycles were recorded in the 5–100 °C range, 0.5 °C min^−1^ using a 300 s equilibration time before each scan. The same method was used for buffer versus buffer scans to obtain the baseline, which was subtracted from sample versus buffer scan to obtain the thermodynamic parameters. No baseline difference was observed before and after the transition, indicating a negligible heat capacity difference between the initial and final states. The experimental enthalpy change, ∆*_exp_**H*°, for the overall unfolding of the DNA structure was estimated by integrating the area under the heat capacity (ΔC_p_°) versus temperature curves and representing the average of at least three different heating experiments. *T*_*m*_ values corresponded to the maximum of each DSC curve. Moreover, DSC curves were fitted by a two-state transition equation according to the van’t Hoff analysis using the NanoAnalyze software (TA instruments, New Castle, DE, USA).

### 3.7. Native Polyacrylamide Gel Electrophoresis Analysis

Native PAGE experiments were performed according to reported procedures [62,70], with minor modifications. Slowly annealed oligonucleotide samples (dissolved at 3 µM concentration) in the selected buffer solutions were loaded on 10% polyacrylamide gels in TBE 1× as running buffer. As additional reference oligonucleotide, the not-annealed (N.A.) V7t1 aptamer dissolved in HEPES/Na^+^ buffer (25 mM HEPES, 150 mM NaCl, pH = 7.4) at 4 μM concentration was also used. All the samples were supplemented with 5% glycerol just before loading and then run, under native conditions, at 80 V at r.t. for 75 min. Gels were stained with a GelGreen solution (supplemented with 0.1 M NaCl) for 30 min, according to manufacture instructions, and finally visualized with a UV transilluminator (BioRad ChemiDoc XRS). Each experiment was performed in duplicate.

### 3.8. Nuclease Stability Assay

The stability in serum of MS3 analogues was determined by gel electrophoresis analysis according to reported procedures [60,61], with minor modifications. Briefly, both aptamers at 40 μM concentrations were incubated in 80% FBS at 37 °C. Then, at fixed times, 3 μL of the samples were collected, mixed with formamide (1:2, *v*/*v*) to immediately quench the enzymatic degradation, heated at 95 °C for 5 min, and finally stored at −20 °C until subsequent analysis. Thereafter, all the samples were supplemented with 5% glycerol immediately before loading and analyzed by gel electrophoresis on 20% denaturing PAGE using 8 M urea in TBE 1X as running buffer. The gels were run at r.t., at constant 200 V for 3 h, then stained with GelGreen nucleic acid stain (supplemented with 0.1 M NaCl) for 30 min and finally visualized with a UV transilluminator (BioRad ChemiDoc XRS, Milan, Italy). The experiment was repeated at least in triplicate. The intensity of the DNA bands on the gel, at each collected time, was then calculated by using the FiJi software and normalized with respect to the untreated oligonucleotide. Percentages of the remaining intact oligonucleotide are reported as mean values ± SD for multiple determinations.

### 3.9. Cell Cultures

SH-SY5Y cells were maintained in RPMI-1640 (Euroclone, Pero, Italy) with 10% fetal bovine serum (FBS; HyClone, Fisher Scientific, Waltham, MA, USA), and 2 mM L-glutamine (Euroclone, Pero, Italy). All cell lines were maintained at 37 °C in a saturated humidity atmosphere containing 95% air and 5% CO_2_.

### 3.10. Fluorescence Microscopy Analysis

To monitor the aptamers’ internalization, FITC-conjugated aptamers were added to the cells in culture medium at 37 °C at different concentrations or for different time periods, as indicated. Then, cells were washed with PBS and fixed with 4% paraformaldehyde (PFA), quenched upon treatment with 50 mM NH_4_Cl. Images were collected using the confocal laser scanning microscope LSM 700 (Carl Zeiss, Jena, Germany) equipped with a Plan Apo 63× oil immersion objective (NA 1.4). Diode lasers at 405 and 488 nm were used as light source; fluorescence emission was revealed by a 505–530 band pass filter for Alexa Fluor 488 and by a 410–460 band pass filter for DAPI. Images were acquired with the confocal pinhole set to one Airy unit using the same setting (laser power, detector gain, threshold of fluorescence intensity) in all experimental conditions.

Image quantitative analysis was performed by using the Zeiss Zen Black software or ImageJ software, as previously described [36,71]. The quantification of mean fluorescence intensities by drawing same-sized regions of interest (ROIs) was measured and corrected for background. For time-course experiments, the mean fluorescence intensities of areas of equal size drawn at the cell surface or inside the cells (close to the nucleus) were measured and corrected for background, as previously described [72].

### 3.11. Drosophila Stocks

Flies were reared on standard corn meal agar with a 12 h on–off light cycle at 25 °C. The fly stocks used in this study were obtained from the Bloomington Stock Center (Bloomington, IN, USA): w*; P{UAS-HTT.128Q.FL}f27b-8765 P{w[+mW.hs]=GawB}elav[C155].

### 3.12. Aptamer Treatment and Crosses

During the tests, flies were reared in vials containing 2 mL of “assay fly food” (AF) (2% agar, 10% powdered yeast, 10% sucrose, 0.1% nipagin) or on the same AF supplemented with different aptamers (MS3 and its truncated MS3-17 and MS3-33 analogues). Proper volumes of each aptamer stock solution in PBS were added onto the surface of the assay fly food and left under gentle agitation for 3 h at room temperature until dried. PBS was supplemented in equal amounts in all the aptamers and in the control food, devoid of aptamer.

Based on the bipartite expression system (UAS)-GAL4 [73], the expression of the UAS-HTT128QFL gene was obtained by crossing females carrying the panneural driver elav-Gal4 to males from the UAS HTT128QFL strain at 28 °C. The parental strains elav/+ and UAS HTT128QFL were used as controls. For all the assays, only one sex was used in the study.

### 3.13. Lifespan Assay

Lifespan assay was carried out as previously described [74]. Briefly, newly emerged adult flies with the desired genotype (P{UASHTT128QFL}f27b/P{w[+mW.hs]=GawB}elav[C155] were collected under cold-induced anesthesia, sorted by sex, grouped into five cohorts of 20 individuals in vials containing 2.0 mL of AF supplemented or not with the different aptamers, and reared at 28 °C. Subsequently, flies were transferred to new vials, with fresh food once every 3 days. The lifespan was measured by recording the number of dead flies at each transfer, until no living flies remained in the vials. Each lifespan measurement used 100 flies, and this was repeated in two independent experiments per treatment. The values obtained were used to calculate the mean lifespan (the mean survival days of all flies for each group) and maximum lifespan (the maximum number of days needed to reach 90% mortality).

### 3.14. Negative Geotaxis Assay

The climbing assay was carried out as previously described [68]. Briefly, 20 sex-matched flies were placed in a graduated empty plastic vial (18 × 2.5 cm) and allowed to recover for 30 min. Negative geotaxis was measured by recording the number of flies that climbed above the 9 cm mark within 20 s after a tap-down of the flies to the bottom of the vial. This assay was repeated for the same group 2 times, allowing for a 1 min rest period between each trial. The number of flies per group that passed the 9 cm mark was recorded as a percentage of total flies. Each trial was performed 3 times at each time point, and the data were expressed as an average of the replicates (*n* = 180).

### 3.15. Statistical Analyses

All quantitative data were presented as the mean ± SD, and the statistical significance was evaluated using one-way ANOVA analysis, followed by Dunnett’s multiple comparisons test for multiple comparisons to determine any statistical differences between groups. Each experiment was performed at least 3 times. Asterisks were used to indicate a significant difference from the controls (* *p* = 0.0208 and **** *p* < 0.0001). One-way ANOVA analysis, followed by the Bonferroni multiple comparison test, was used for fluorescence-based experiments; *p*-values are indicated in the figures.

All the data were analyzed with the GraphPad Prism 9 statistical software package (GraphPad, La Jolla, CA, USA).

## 4. Conclusions

A large number of G-quadruplex-forming aptamers has been discovered and found to display significant activity in several innovative therapeutic and diagnostic strategies [21,22]. In the context of anti-HD research, a set of G-quadruplex-forming aptamers was recently identified to selectively bind a mutant huntingtin protein, mHTT, carrying a 78-residue polyglutamine tract expansion and inhibit its activity [34]. We thus performed a detailed physicochemical and biological characterization on the best aptamer in this series, named MS3, proved to adopt a very stable parallel G-quadruplex folding, show high resistance to nucleases, and be rapidly internalized in both non-neuronal HeLa and neuronal SH-SY5Y cells in a dose-dependent manner even without transfecting agents [36]. Then, using a well-established Drosophila melanogaster model for HD (Q128HD-FL), which expresses the mutated form of human huntingtin mHTT, MS3 treatment allowed for markedly improving the neuronal function of the transgenic flies, clearly proving the in vivo efficacy of this aptamer.

Aiming at further optimizing MS3, we here investigated two novel truncated variants of this aptamer, designed to reduce its intrinsic tendency to form large multimeric G-quadruplex aggregates: MS3-33, a 33-mer lacking the 5′-terminal GGG tract of MS3 sequence, and MS3-17, a 17-mer containing ca. half the sequence of MS3-33. Both oligonucleotides were analyzed by UV, CD, and DSC in pseudo-physiological buffers, overall demonstrating similar and, in some cases, improved properties compared with the parent aptamer. Gel electrophoresis analysis under native conditions showed that the MS3 analogues had somehow lower heterogeneity than the parent aptamer, even though only in the case of MS3-17 large aggregates and/or higher order G-quadruplex structures were not formed. This favorable condition could, however, explain the lower resistance to nucleases of MS3-17 compared with MS3-33 and, even more, with the parent MS3 aptamer. Anyway, from a general point of view, if unmodified oligonucleotides are completely degraded in FBS in a few minutes, the here investigated oligonucleotides proved to resist nucleases for a few hours similar to other G-quadruplex-forming aptamers [75]. Confocal microscope analysis on fluorescein-labelled derivatives of MS3-33 and MS3-17 confirmed the ability of these MS3 analogues to be rapidly and dose-dependently taken up by SH-SY5Y cells even in the absence of transfecting agents, not causing evident cytotoxicity. Notably, in the in vivo tests on transgenic flies, both MS3-derived analogues always gave better results than MS3, definitively proving an improved bioactivity of these shorter oligonucleotides compared with their parent aptamer.

This large multidisciplinary work demonstrated the efficacy of the novel truncated aptamers both in vitro and in vivo. Of particular interest is MS3-17, a much shorter and easy-to-handle analogue than MS3, which did not form multimeric G-quadruplex-based aggregates in pseudo-physiological buffer solutions but maintained satisfactory chemical and enzymatic stability and displayed more rapid cell uptake and higher cellular persistence. Its very promising biological activity in a Drosophila model denotes high potential in the anti-HD therapeutic approaches based on specifically targeting mutant huntingtin.

## Figures and Tables

**Figure 1 ijms-23-12412-f001:**
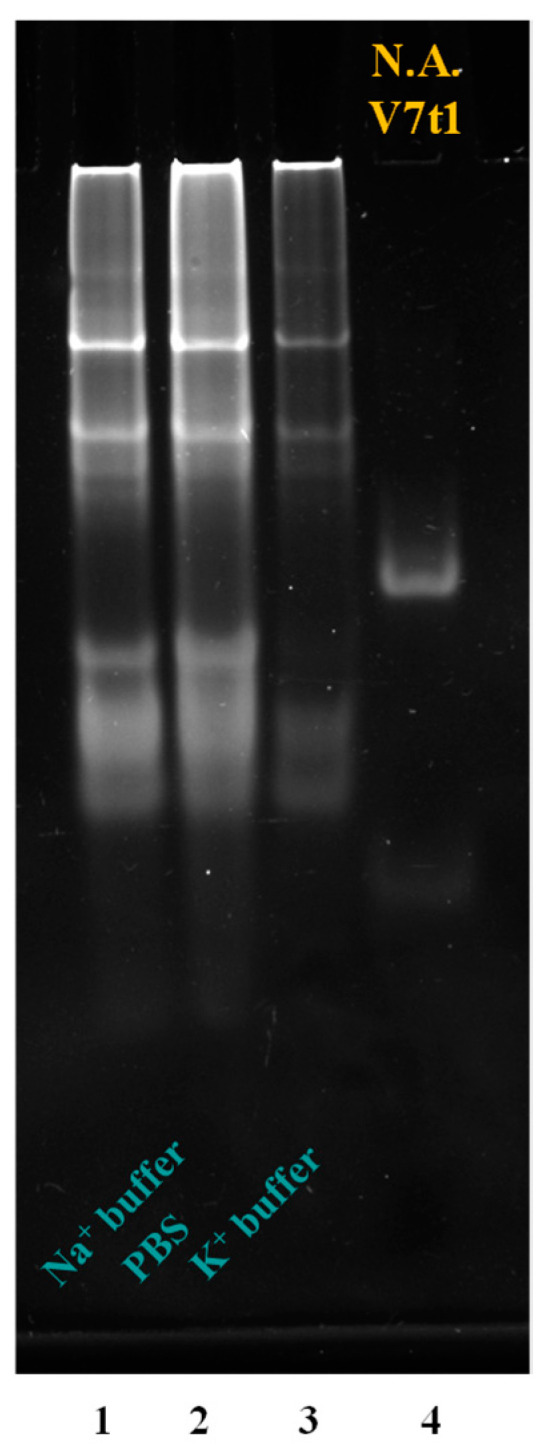
Representative 10% polyacrylamide gel electrophoresis under native conditions of MS3, at 3 μM concentration in the indicated buffer solutions, run at constant 80 V at r.t. for 75 min in TBE 1× buffer. The not-annealed (N.A.) V7t1 aptamer dissolved in HEPES/Na^+^ buffer (25 mM HEPES, 150 mM NaCl, pH = 7.4) at 4 μM concentration was used as reference oligonucleotide. Lane 1: MS3 in Na^+^ buffer; lane 2: MS3 in PBS; lane 3: MS3 in K^+^ buffer; lane 4: V7t1.

**Figure 2 ijms-23-12412-f002:**
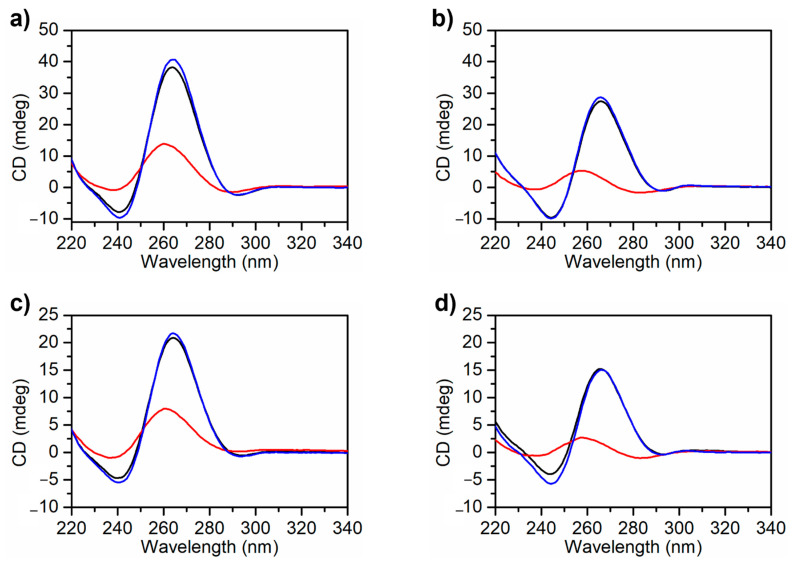
CD analysis of MS3 analogues: CD spectra of MS3-33 (**a**) and MS3-17 (**c**) in PBS buffer (137 mM NaCl, 2.7 mM KCl, 10 mM NaH_2_PO_4_/Na_2_HPO_4_, 1.8 mM KH_2_PO_4_/K_2_HPO_4_, pH = 7.4) at 5 °C before (black line), after melting/cooling (blue line), and at 100 °C (red line); CD spectra of MS3-33 (**b**) and MS3-17 (**d**) in the Na^+^ buffer (10 mM NaH_2_PO_4_/Na_2_HPO_4_, 90 mM NaCl solution, pH = 7.0) at 5 °C before (black line), after melting/cooling (blue line), and at 100 °C (red line).

**Figure 3 ijms-23-12412-f003:**
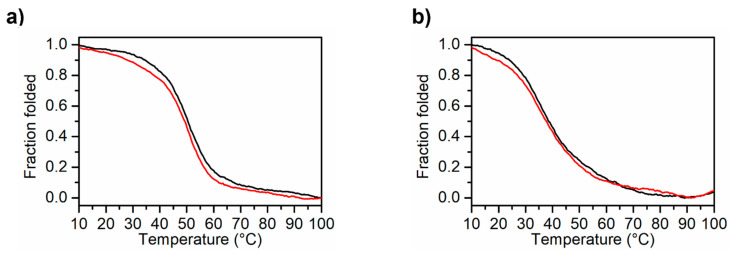
CD melting experiments of MS3 analogues recorded at a scan rate of 0.5 °C min^−1^: CD melting (black line) and cooling (red line) profiles of MS3-33 (**a**) and MS3-17 (**b**) in the Na^+^ buffer (10 mM NaH_2_PO_4_/Na_2_HPO_4_, 90 mM NaCl solution, pH = 7.0).

**Figure 4 ijms-23-12412-f004:**
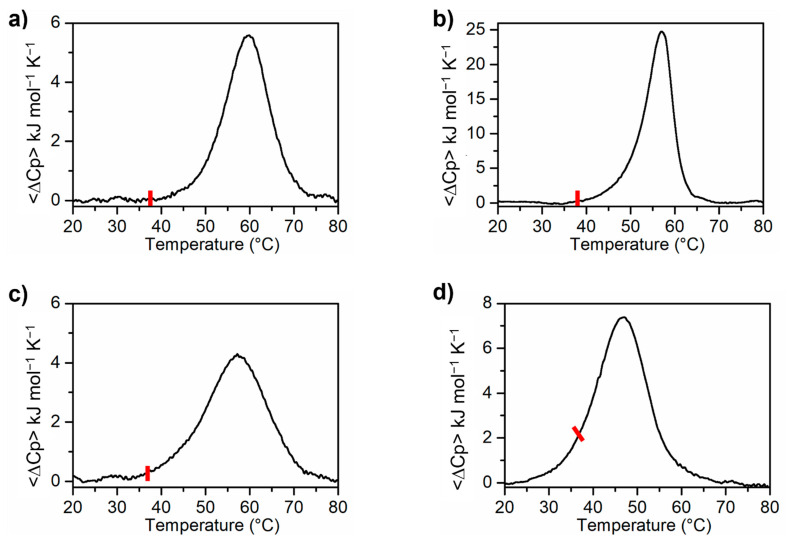
Experimental DSC profiles for MS3-33 (**a**) and MS3-17 (**c**) in PBS buffer (137 mM NaCl, 2.7 mM KCl, 10 mM NaH_2_PO_4_/Na_2_HPO_4_, 1.8 mM KH_2_PO_4_/K_2_HPO_4_, pH = 7.4); DSC profiles for MS3-33 (**b**) and MS3-17 (**d**) in the Na^+^ buffer (10 mM NaH_2_PO_4_/Na_2_HPO_4_, 90 mM NaCl solution, pH = 7.0). The red bar indicates the physiological temperature.

**Figure 5 ijms-23-12412-f005:**
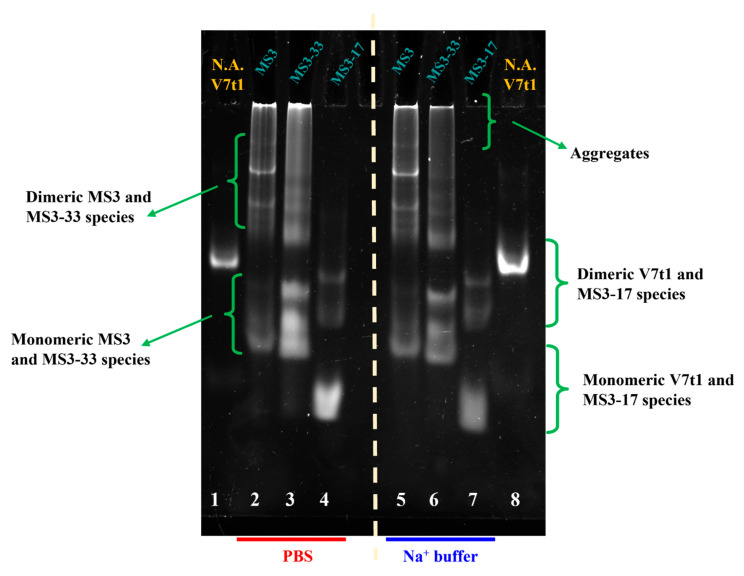
Representative 10% polyacrylamide gel electrophoresis under native conditions of MS3, MS3-33, and MS3-17 samples at 3 μM concentration in the selected PBS (lanes 2–4) and Na^+^ buffer (lanes 5–7) solutions, run at constant 80 V at r.t. for 75 min in TBE 1× buffer. The not-annealed (N.A.) V7t1 aptamer dissolved in HEPES/Na^+^ buffer (25 mM HEPES, 150 mM NaCl, pH = 7.4) at 4 μM concentration was used as a control oligonucleotide (lanes 1 and 8). Lane 1: V7t1; lane 2: MS3; lane 3: MS3-33; lane 4: MS3-17; lane 5: MS3; lane 6: MS3-33; lane 7: MS3-17; lane 8: V7t1.

**Figure 6 ijms-23-12412-f006:**
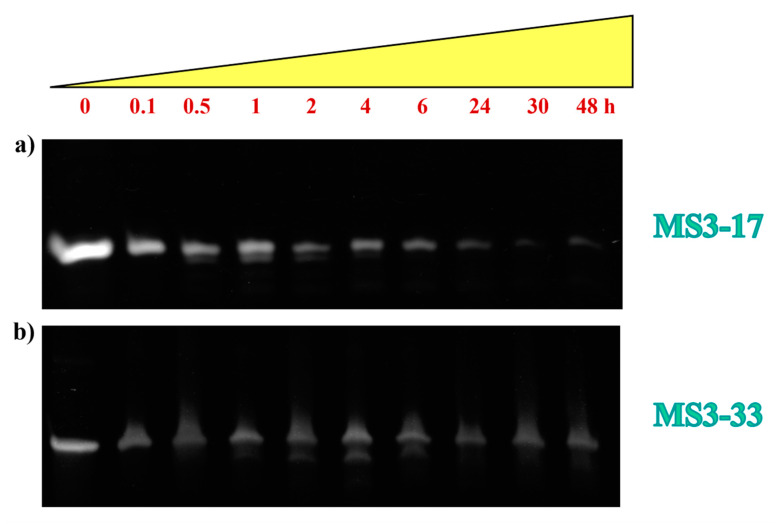
Enzymatic resistance experiments performed on MS3-17 (**a**) and MS3-33 (**b**) incubated in 80% fetal bovine serum (FBS) as monitored by 20% denaturing polyacrylamide gel electrophoresis up to 48 h (time points: 0, 0.1, 0.5, 1, 2, 4, 6, 24, 30, and 48 h). For each compound, a representative 20% denaturing PAGE (8 M urea) is reported. Gels were run at a constant 200 V at r.t. for 3 h in TBE 1X as running buffer.

**Figure 7 ijms-23-12412-f007:**
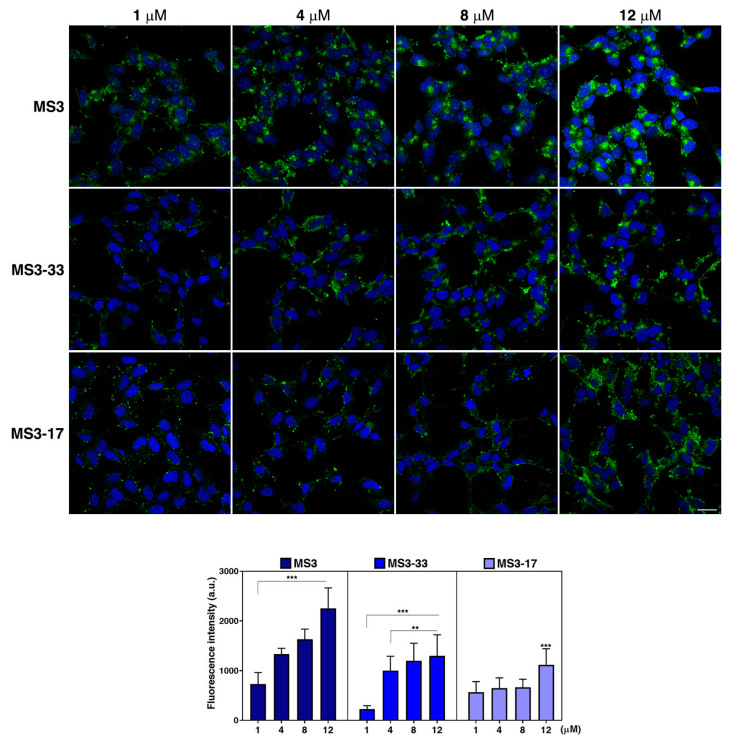
Uptake of FITC-conjugated MS3 and its truncated analogues in neuronal cells. SH-SY5Y cells were incubated with the three FITC-conjugated aptamers (green) for 24 h at different concentrations, as indicated. Then, cells were fixed, and nuclei were stained with DAPI (blue). Images were acquired with a confocal microscope. Scale bars, 6 μM. Mean fluorescence intensity of three independent experiments is shown, *n* > 50 cells; error bars, mean ± SD. *** *p* < 0.001, ** *p* < 0.01, Bonferroni test after significant ANOVA.

**Figure 8 ijms-23-12412-f008:**
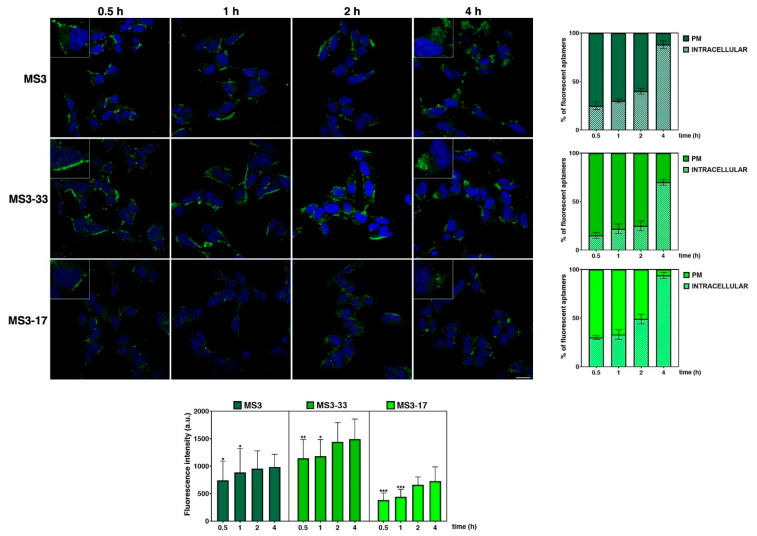
Time-course analysis of cell uptake of MS3 and its analogues. SH-SY5Y cells were incubated with a 4 µM solution of FITC-conjugated aptamers (green) at the indicated time points and fixed; then, nuclei were stained with DAPI (blue). Images were acquired with a confocal microscope. Higher magnification pictures in the insets. Scale bar, 6 µm. Mean fluorescence intensity of two independent experiments is shown, *n* > 50 cells; error bars, mean ± SD. *** *p* < 0.001, ** *p* < 0.01, * *p* < 0.05, Bonferroni test after significant ANOVA. Mean fluorescence intensities at the plasma membrane (PM) or inside the cell (intracellular) were measured at the different times and expressed as percentage of total fluorescence; *n* > 30 cells.

**Figure 9 ijms-23-12412-f009:**
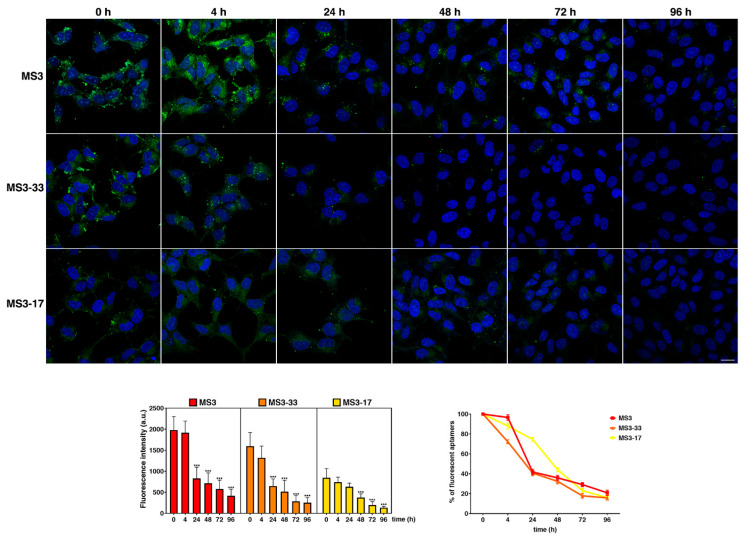
Analysis of the biological stability in SH-SY5Y cells of MS3 and its analogues by time-course experiments. SH-SY5Y cells were incubated for 24 h with FITC-conjugated aptamers (green). After the removal of the residual aptamers by extensive washings, cells were incubated in a culture medium for the different indicated times. Nuclei were stained with DAPI. Images were acquired with a confocal microscope. Scale bar, 6 µm. Mean fluorescence intensity was measured at the different time points and expressed as an arbitrary unit (a.u.; left graph) or as the percentage of fluorescence detected at time 0 (set to 100%; right graph); error bars, mean ± SD; *n* > 40 cells. *** *p* < 0.001, Bonferroni test after significant ANOVA. Note that there is a statistically significant difference between the 72 and 96 h time points for MS3 but not for MS3-33 and MS3-17 (*p*-value < 0.001, Bonferroni test after significant ANOVA).

**Figure 10 ijms-23-12412-f010:**
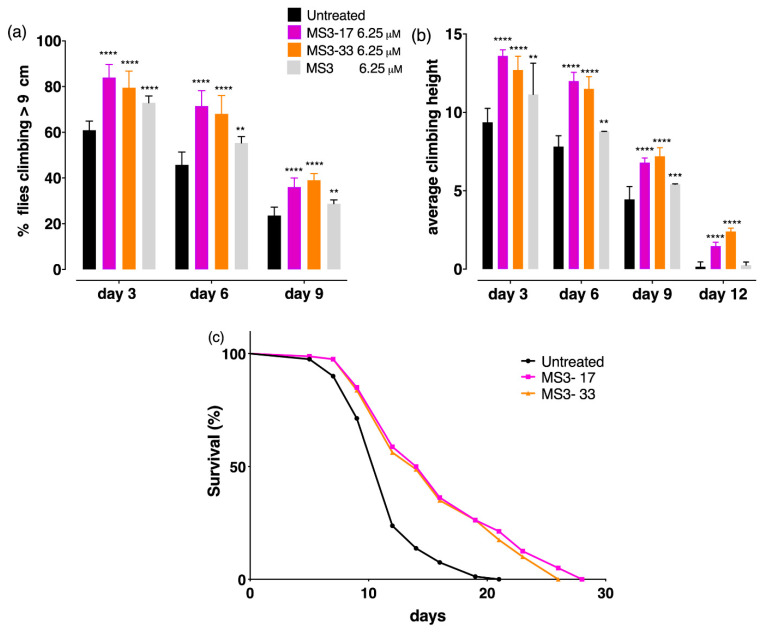
MS3-derived aptamers’ suppressed HD symptoms in a Drosophila model. Administration of MS3 and of its two analogues (MS3-17 and MS3-33) during adult period suppressed motor dysfunction in flies expressing HttFLQ128. Comparison was made between untreated and treated flies of the same age. (**a**) Climbing assay. The percentage of flies that, after the MS3 analogues’ (MS3-17 and MS3-33) chronic treatment throughout adulthood, could reach the 9 cm target was higher than both untreated and MS3-treated sibling flies at each point examined. (**b**) The average climbing height of MS3 analogues was significantly higher compared with MS3-treated flies also on day 12 after eclosion. (**c**) The survival curves of both treated groups declined slower in comparison with the curve of the untreated flies that quickly dropped on day 7, and the mean and maximum lifespan were significantly extended compared with untreated flies. **** *p* < 0.0001; *** *p* = 0.0005; ** *p* = 0.023, compared with untreated flies. Data represent mean ± SD.

**Table 1 ijms-23-12412-t001:** Apparent melting temperature values obtained for the unfolding (*T_m_*) and refolding (*T_c_*) of MS3-33 and MS3-17 analogues by UV-monitored thermal experiments at 295 nm in the selected saline conditions. The error on *T_m_* and *T_c_* determination is ±1.0 °C.

	Sample	*T*_*m*_ (°C)	*T*_*c*_ (°C)
**PBS**	MS3-33	63	61
	MS3-17	53	52
**Na^+^ buffer**	MS3-33	59	56
	MS3-17	46	51

**Table 2 ijms-23-12412-t002:** Apparent melting temperature values (*T_m_*) obtained for the unfolding of MS3-33 and MS3-17 analogues in comparison with MS3. The error on *T_m_* determination is ±1.0 °C.

	Sample	*T_m_* (°C)
**PBS**	MS3-33	62
	MS3-17	55
	MS3	N.D. ^a^
**Na^+^ buffer**	MS3-33	50
	MS3-17	37
	MS3	50

^a^ Not determined.

**Table 3 ijms-23-12412-t003:** Thermodynamic parameters obtained by the differential scanning calorimetry profiles. The error on *T_m_* determination is ±1.0 °C and on Δ*H°* is ±10%.

	Samples	*T_m_* (°C)	Δ*_exp_H°* (kJ mol^−1^)	Δ*_vH_H°* (kJ mol^−1^)
**PBS**	MS3-33	60	68	285
	MS3-17	57	77	197
**Na^+^ buffer**	MS3-33	57	224	270
	MS3-17	47	134	201

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
