# Peer review of "Truncated Analogues of a G-Quadruplex-Forming Aptamer Targeting Mutant Huntingtin: Shorter Is Better!"

_ijms, 2022, doi:10.3390/ijms232012412_

Round 1
Reviewer 1 Report
The menuscript is very interesting and sel-sustained. I suggest the acception for further publication. My single comment is about the thermodynamic parameters calculated from the melting curves.
The thermodynamic parameters can’t be calculated for non-equilibrium processes. Some UV-melting and CD-melting curves do not coincide with folding curves, so the thermodynamic parameters can’t be estimated (Fig. S3 and S6). The additional decrease in the melting rate either alteration of folding process can be used to gain the equilibrium state.
Author Response
We agree with this Reviewer that UV-melting measurements show a significant hysteresis for MS3 in the Na+ buffer (Fig. S3) and we commented this in the text (see paragraph 2.2). We calculated van ’t Hoff enthalpies by CD-melting data obtained at 0.5 °C/min. In this case, we observed a slight hysteresis only for MS3-33 in PBS (Fig. S6). We are well aware that van’ t Hoff enthalpy values are model-dependent and require an equilibrium between folded and unfolded species. We commented this point in the text and for this reason we performed calorimetric analyses, where model-independent enthalpies are directly derived by DSC curves. Following this Reviewer’s suggestion and for better clarity, we removed van’ t Hoff enthalpies, calculated by CD-melting measurements, from previous Table 1 (now Table 2 in the revised version) and the related comment in the text (paragraph 2.3). We also added Tm values of MS3 for comparison, as required by Reviewer #2. However, we calculated van’ t Hoff enthalpies by DSC curves, inserted the thus obtained enthalpy values in Table 3 and discussed the results in the text (pag.7).
Reviewer 2 Report
In the manuscript entitled “Truncated analogues of a G-quadruplex-forming aptamer targeting mutant huntingtin: shorter is better!”, Riccardi and coworkers investigated the physiochemical and biological properties of two new G4 forming aptamers analogues of the aptamer MS3 which was previously reported to bind and regulate mutant huntingtin protein involved in the pathology of Huntington’s disease. The authors extensively characterized the G4 folding properties and structural characteristics of the newly reported aptamers and they additionally tested them in neuronal cells and in an in vivo Drosophila model of Huntington’s disease. The topic presented in this manuscript is interesting and the data confirms the therapeutic potential of G4-forming aptamers in targeting proteins involved in pathological processes and diseases. Overall, the manuscript is well written and the results support the conclusion stated by the authors. However, I suggest the following improvements to the prior to publication:
Major:
· Some experiments are lacking essential controls (i.e. oligos with mutations in the G4 sequence that doesn’t allow G4 formation but maintain similar length and base composition).
· No binding studies were performed to explore the binding properties of the two new aptamers compared to MS3.
· Add MS3 data to table 1 to allow easier comparison
· Page 11, lines 331-333. I can only see the additional band for MS3-17 at 1hr. Is there a way to improve the image quality?
· Page 12, lines 361-364. Cytotoxicity shroud be confirmed with other assays (i.e. MTT or trypan blue exclusion assay)
· Figure 7: MS3-33 and MS3-7 show very poor dose-dependent uptake. Please comment.
· Page 14, Lines 401-409. The authors state that a “valuable signal is detectable until 72hrs” but based on the graph showing the % of fluorescence detected, the signal at 72hrs is similar to the one detected at 96hrs. Please comment and clarify this observation.
Minor:
· I would add a figure to show the differences between MS3, MS3-33 and MS3-17 in additional to the explanation in the text.
· Section 2.2 is quite long and needs to be improved for clarity
· Is suggest to add at least one figure/table from this section 2.2 to the main manuscript (and not only Supplementary info). For example, a table with the UV-derived data would make easier the comparison with the CD-derived data
Author Response
In the manuscript entitled “Truncated analogues of a G-quadruplex-forming aptamer targeting mutant huntingtin: shorter is better!”, Riccardi and coworkers investigated the physiochemical and biological properties of two new G4 forming aptamers analogues of the aptamer MS3 which was previously reported to bind and regulate mutant huntingtin protein involved in the pathology of Huntington’s disease. The authors extensively characterized the G4 folding properties and structural characteristics of the newly reported aptamers and they additionally tested them in neuronal cells and in an in vivo Drosophila model of Huntington’s disease. The topic presented in this manuscript is interesting and the data confirm the therapeutic potential of G4-forming aptamers in targeting proteins involved in pathological processes and diseases. Overall, the manuscript is well written and the results support the conclusion stated by the authors. However, I suggest the following improvements to the prior to publication:
Major:
- Some experiments are lacking essential controls (i.e. oligos with mutations in the G4 sequence that doesn’t allow G4 formation but maintain similar length and base composition).
ANSWER: We thank this Reviewer for the valuable comment, which allows us better explaining this important issue. As well clarified in the Introduction, the selected sequences MS3-33 and MS3-17 were selected on the basis of a rational truncation of MS3, an aptamer which was discovered by Shin et al. (ref. 34) and was recently studied in more detail by us (ref. 36). We think that experiments with a scrambled non G4-forming sequence are not essential at this stage because the procedure used in ref. 34 selected as best aptamers only the oligomers which are able to adopt G4 structures, i.e. MS3 along with other G-rich oligonucleotides (MS1, MS2 and MS4), “among 3,416 aptamer probes in a single strand DNA format”. All the data shown in refs 34, 36 and in the present work clearly confirm that a G4 conformation is required in the biologically active species. Furthermore, in our previous work (ref. 36) we analyzed a scrambled sequence of MS3 – i.e. an oligonucleotide of the same length and similar base composition as MS3 but unable to form G-quadruplex structures - as a negative control in cellular studies, and the obtained results further validated MS3 as a good starting aptamer for further optimization studies.
- No binding studies were performed to explore the binding properties of the two new aptamers compared to MS3.
ANSWER: Our studies are addressed to find the best aptamers biologically active on in vitro and in vivo models to be advanced to clinical trials. We fully agree that an in-depth structural investigation involving binding studies with mHTT protein would give very precious information for the aptamer optimization. Unfortunately, binding studies in vitro by biophysical methodologies are not so trivial due to the difficulties in mHTT protein expression and purification and currently we are absolutely unable to perform them. However, we are firmly confident that the experiments we here presented in cell models, and particularly those in vivo on the Drosophila Huntington’s disease model, provide conclusive data on the improved properties of truncated MS3 aptamers with respect to unmodified MS3.
- Add MS3 data to table 1 to allow easier comparison
ANSWER: Following this Reviewer’s request, we added MS3 data in Table 1 -which in the revised version is Table 2 - for a better comparison with its truncated analogues.
- Page 11, lines 331-333. I can only see the additional band for MS3-17 at 1 hr. Is there a way to improve the image quality?
ANSWER: Following this Reviewer’s request, in the revised version of the manuscript we replaced the original image with one with improved resolution (Figure 6).
- Page 12, lines 361-364. Cytotoxicity should be confirmed with other assays (i.e. MTT or trypan blue exclusion assay)
ANSWER: We thank this Reviewer for the valuable suggestion. According to this request, we performed trypan blue exclusion and MTT assays (see new Supplementary Figure S10). No difference between untreated and treated cells was observed at the highest aptamer concentrations tested (12 mM), confirming that the treatment with aptamers is not cytotoxic under the studies conditions.
- Figure 7: MS3-33 and MS3-17 show very poor dose-dependent uptake. Please comment.
ANSWER: We deeply thank this Reviewer for raising this issue and allowing us better explaining this point. In this new version of our manuscript, we have better clarified that the two analogues have a different behavior in terms of dose-dependent cell uptake, also discussing possible reasons to explain these experimental results (see page 12).
- Page 14, Lines 401-409. The authors state that a “valuable signal is detectable until 72 hrs” but based on the graph showing the % of fluorescence detected, the signal at 72 hrs is similar to the one detected at 96 hrs. Please comment and clarify this observation.
ANSWER: Also in this case we are very grateful to this Reviewer for her/his precious comment. Firstly, we apologize because we realized that in the submission process the images showing immunofluorescence assays appeared with low contrast (we provided tif images as separate files to the Editor) which could be misleading or difficult to interpret. To avoid possible misinterpretations, we have provided pictures with a better contrast in the submission of the revised manuscript.
The intensity of fluorescence signal decreases over time and is very low after 72 and 96 h for all aptamers; we have better clarified this in the text (see page 14). Moreover, following this Reviewer’s suggestion, we extended the statistical analyses comparing all the timepoints among them. In this analysis, we observed a statistical significance difference between 72 and 96 h only for MS3 but not for MS3-33 and MS3-17. We have provided the results of this analysis in the caption to Figure 9.
Minor:
- I would add a figure to show the differences between MS3, MS3-33 and MS3-17 in additional to the explanation in the text.
ANSWER: We thank this Reviewer for her/his valuable suggestion. Accordingly, we added a new figure in the Supplementary Materials (Figure S8), showing the overlapped CD spectra of the three aptamers to highlight their conformational differences.
- Section 2.2 is quite long and needs to be improved for clarity
ANSWER: We thank this Reviewer for her/his valuable suggestion. Accordingly, we reduced and simplified the description of this section.
- I suggest to add at least one figure/table from this section 2.2 to the main manuscript (and not only Supplementary info). For example, a table with the UV-derived data would make easier the comparison with the CD-derived data
ANSWER: Following this Reviewer’s suggestion, in the revised version of the manuscript we moved the Table with the UV-derived data (previously Table S2) in the main manuscript (now Table 1).
Reviewer 3 Report
This paper by Riccardi et al. reports the biophysical characterization and biological evaluation of two truncated MS3-based aptamers against the mutated huntingtin protein. For both truncated aptamers, the authors observed improved performance compared to the parent aptamer MS3, which they attribute to reduced formation of larger aggregates.
This is a nice and through work of high importance, the experiments appear sound and the conclusions well supported by the data. There is only one minor issue that should be addressed before publication in IJMS:
It was recently demonstrated by Koga et al. (https://doi.org/10.1021/acs.biomac.2c00282) that cellular uptake of DNA nanostructures is modulated by the type of attached fluorophores. Furthermore, it was found that this modulation depends on the cell line and the type and modification of the DNA nanostructue. Therefore, I suggest the authors add a brief discussion of these issues to section 2.7 and mention that the uptake of non-labelled aptamers may differ from the presented data.
Author Response
We deeply thank this Reviewer for the precious comment. In the revised version of our manuscript, we have discussed this issue and commented our data in the light of the results reported in the article by Koga et al. (page12).